# USING APPROXIMATE MODELS FOR EFFICIENT EXPLORATION IN REINFORCEMENT LEARNING

## ABSTRACT

In model-based reinforcement learning, an agent uses a learned model of environment dynamics to improve a policy. Using a learned model of the environment to select actions has many benefits. It can be used to generate experience for learning a policy or simulate potential outcomes in planning. It allows flexible adaptation to new tasks and goals without having to relearn the underlying fundamentals of the environment from scratch. These sample efficiency and generalisation gains from model use are restricted by the model's accuracy. An imperfect model can lead to failure if trusted by the agent in regions of the state space where predictions are inaccurate. It is well-documented in cognitive and developmental psychology that humans use approximate intuitive models of physics when navigating the world in everyday scenarios. These intuitive models, despite being imperfect, enable humans to reason flexibly about abstract physical concepts (for example, gravity, collisions and friction), and to apply these concepts to solve novel problems without having to relearn them from scratch. In other words, humans efficiently make use of imperfect models. In this paper, we learn dynamics models for intuitive physics tasks using graph neural networks that explicitly incorporate the abstract structure of objects, relations and events in their design. We demonstrate that these learned models can flexibly generalise to unseen tasks and, despite being imperfect, can improve the sample efficiency of policy learning through guiding exploration to useful regions of the state and action space.

## 1 INTRODUCTION

In model-based reinforcement learning, the agent possesses a model of the dynamics of the environment, and learns from the experience generated by this model (Sutton & Barto, 2018). The model is either learned or given. Using the experience generated from a model to improve a policy and/or value function is often referred to as planning. Generating experience or transitions from a model and learning from this can significantly improve the sample efficiency of learning, since fewer direct interactions with the environment are required to generate experience. Using a model can enable flexible adaptation to new tasks and goals without having to relearn the underlying fundamentals of the environment from scratch (Lake et al., 2016). These sample efficiency and generalisation gains from model use are restricted by the model's accuracy. An imperfect model can lead to failure if trusted by the agent in regions of the state space where predictions are inaccurate (Abbas et al., 2021). Relating to the use of imperfect or approximate models in the model-based reinforcement learning literature, there has been a focus on (1) mitigating the impact of model inaccuracies on policy learning by restricting rollout length or relying less on uncertain predictions (Janner et al., 2019; Abbas et al., 2021); and (2) learning and using simplified models that abstract away details that are irrelevant for value-based planning (Grimm et al., 2020; 2021; Schrittwieser et al., 2019), or, more generally, learning the model in a way that links it to its use in value-based planning (Saleh et al., 2022).

It is well-documented in cognitive and developmental psychology that humans use approximate intuitive models of physics when navigating the world in everyday scenarios (Piloto et al., 2022). These intuitive models enable humans to reason flexibly about abstract physical concepts such as gravity, collisions and friction. Humans demonstrate the capacity to construct rich action priors based on their intuitive models of physics to flexibly solve physical reasoning problems (Allen et al., 2020). These action priors can be used to guide exploration through biasing and restricting the search over

actions to those that have been previously useful, thereby, speeding up learning (Rosman & Ramamoorthy, 2015). Taking inspiration from this, we propose using learned dynamics models to guide exploration towards rewarding regions of the action space to accelerate learning. Efficient exploration is a major challenge for reinforcement learning algorithms, especially in environments with continuous action spaces and where a sequence of precise actions have to be chained together in order to receive some sparse reward signal from the environment (Ecoffet et al., 2021). By using approximate and imperfect models to guide exploration only and not generate experience or transitions from which the reinforcement learning algorithm directly learns, we mitigate the impact of model error on stable policy learning, while still benefiting from the information contained in the predictions of these models, however uncertain.

We make the following contributions:

- We learn dynamics models for intuitive physics tasks that leverage the abstract structure of objects, relations and events to flexibly generalise across tasks. To do this, we learn the models using graph neural networks based on the Interaction Network (Battaglia et al., 2016), that rely on the assumptions of distinct edge update functions for unique pairs of object types, dynamic edge activations and relative information.

- Using the learned approximate dynamics models, we compute the statistics of action distributions identified as rewarding by the model, and show that sampling from these distributions significantly improves the frequency of reward signals obtained from the environment.

- We incorporate the model-identified rewarding action distributions in an epsilon-greedy exploration strategy for reinforcement learning that demonstrates accelerated convergence to optimal policies.

## 2 BACKGROUND

We model the agent's interaction with the environment as a Markov Decision Process, represented by a tuple $\langle \mathcal{S}, \mathcal{A}, \mathcal{P}, \mathcal{R}, \gamma \rangle$ of states, actions, transition probability function, reward function and discount factor. At each time step $t$, the agent observes the environment state $s_t$, and then selects an action $a_t$. At the next time step, the agent receives reward $r(s_t, a_t)$ and transitions to the next state $s_{t+1}$ according to some dynamics transition function $f : \mathcal{S} \times \mathcal{A} \rightarrow \mathcal{S}$. The goal of reinforcement learning is for an agent to learn a policy $\pi(a_t|s_t)$, a mapping from states to actions, that maximises the expected sum of future rewards that can be achieved from the current state $\mathbb{E}_\pi[\sum_{i=0}^{\infty} \gamma^i r(S_{t+i}, A_{t+i})|S_t = s]$.

In model-free reinforcement learning, the dynamics function $f$ and reward function $r$ are unknown and the agent learns only from direct interaction with the environment. In model-based reinforcement learning, the agent has a model of the environment, which represents the dynamics function $f$ and reward function $r$, and learns from the experience generated by this model (Sutton & Barto, 2018). The model is either learned or given. In this work, we assume that the agent has access to the underlying reward function $r$ and we focus on learning the dynamics function $f$.

### 2.1 GRAPH NEURAL NETWORKS FOR PHYSICS PREDICTION

Graph neural networks (GNNs) have demonstrated success in flexible physics prediction, with the capacity to generalise across variable object counts and scene configurations (Battaglia et al., 2016; Chang et al., 2017; Sanchez-Gonzalez et al., 2020). This success is attributable to their use of strong relational inductive biases - assumptions about the relationships and interactions among entities (Battaglia et al., 2018). The Interaction Network (Battaglia et al., 2016) assumes pairwise relations between entities, enabling scaling to variable object counts. It represents objects as the nodes of a graph and their relations as edges. The Interaction Network is an example of a Message Passing Neural Network (Gilmer et al., 2017), which computes the updated node fetaures $x_i'$ as follows:

$$x_i' = \psi(x_i, \sum_{j \in \mathcal{N}(i)} \phi(x_i, x_j, d_{j,i})), \tag{1}$$

where $x_i \in \mathbb{R}^F$ denotes the features of node $i$, $d_{j,i}$ denotes the (optional) edge features from sender node $j$ to receiver node $i$, $\psi$ is the per-node differentiable update function, and $\phi$ is the per-edge

differentiable update function that computes the effect of the interaction between receiver node $i$ and sender node $j$ (usually instantiated as multilayer perceptrons). Note that the GNN is permutation-invariant because the per-edge update function $\phi$ is symmetrically applied to all interactions; and the interaction effects are aggregated using a permutation-invariant operation, in this case, summation. A larger, more complex system can be represented by a graph structure $G = \langle \mathcal{O}, \mathcal{R} \rangle$, where nodes $\mathcal{O}$ represent objects and edges $\mathcal{R}$ represent their pairwise relations. The per-node and per-edge update rules in 1 are applied across the full graph to update the features of receiver nodes for the next step.

## 2.2 Deep Deterministic Policy Gradients (DDPG)

DDPG (Lillicrap et al., 2019) is an off-policy reinforcement learning algorithm that uses the Bellman equation to concurrently learn an action-value or $Q$-function and a deterministic policy $\mu(s)$. DDPG is adapted for continuous action spaces by using the following approximation $\max_{\mathrm{a}} Q(s, a) \approx Q(s, \mu(s))$. Since the action space is continuous, the optimal $Q^*(s, a)$ is differentiable with respect to the action. Since the policies learned are deterministic, noise is added to the actions sampled from the policy at training time (usually using an Ornstein-Uhlenbeck (Uhlenbeck & Ornstein, 1930) or Gaussian noise process) for exploration.

## 3 Related Work

**Approximate models and model-based reinforcement learning.** Recently, there has been an increased focus in the model-based reinforcement learning literature on learning useful models as opposed to accurate models (Saleh et al., 2022). Saleh et al. (2022) motivate that a model that accurately predicts environment dynamics may not necessarily be the most useful to a learner and instead focus on learning planning models that can be used to improve value function learning as a meta objective. Other works have investigated approaches for reducing the detrimental impacts of inaccurate models on learning. Abbas et al. (2021) propose an approach for estimating model uncertainty and use this uncertainty measure to limit the impact of planning in regions of the state space where models are inaccurate. Limiting the length of model rollouts to reduce compounding model error is a widely adopted technique in the literature (Janner et al., 2019; Holland et al., 2019). The value equivalence principle (Grimm et al., 2020) places emphasis on accurately predicting returns and not environment transitions. We propose that approximate models can be useful by directing exploration towards useful regions of the action space in settings with large or continuous action spaces and where rewards are sparse.

**Intuitive physics.** It is well-established in cognitive and developmental psychology that people use approximate and abstract models of physics when interacting with the physical world (Spelke, 1990; 2003; Gerstenberg & Tenenbaum, 2020). Much of the work on learning models for physical intuition focuses on the specialised tasks of block stacking and stability prediction (Battaglia et al., 2013; Groth et al., 2018; Lerer et al., 2016). Girdhar et al. (2021) and Allen et al. (2020) focus on intuitive physics puzzles. However, the tasks they investigate are one-step and do not involve the sequential decision-making typical of traditional reinforcement learning tasks. Allen et al. (2020) also do not learn a dynamics model, but use a ground truth simulator with added noise for their experiments.

**Action priors and hard exploration.** Exploration can be difficult in environments with large or continuous action spaces and where a sequence of precise actions have to be chained together in order to receive some sparse reward signal from the environment. To enable more effective exploration, Ecoffet et al. (2021) store states visited during exploration, revisit promising states and explore starting from the revisited states in a model-free way. This mitigates the issues of the agent forgetting how to reach previously visited states and not first returning to a state before exploring from it. Action priors can be used to guide exploration through biasing and restricting the search over actions to those that have been previously useful, thereby, speeding up learning (Rosman & Ramamoorthy, 2015). We propose using a learned model to explore from promising states and using the statistics of promising actions identified by the model to compute cluster distributions of rewarding actions that can be used to guide exploration.

**Graph neural networks for simulating physics**. Battaglia et al. (2016) assume a fixed set of nodes and edges for training, with only node features varying over time. Sanchez-Gonzalez et al. (2020) simulate complex physics (fluids, rigid solids, deformable materials) assuming a fixed set of nodes and dynamic edge structures using a connectivity radius - that is, edges are constructed between nodes within some specified radius. Saxena & Kroemer (2022) learn time-varying sparse graph

structures to scale to tasks with a larger number of 'distractorqnodes (nodes that are not involved in any interactions). Edge activations refer to an edge between the relevant nodes existing at a discrete time point. They focus on robotic gripping tasks where the assumption of longer contact durations is relevant and use a temporal Gated Recurrent Unit to learn edge activations. In training, they use a fixed set of nodes. In contrast, our training dataset is noisy with nodes randomly added at regular intervals.

## 4 LEARNING AND USING APPROXIMATE MODELS TO GUIDE EXPLORATION IN INTUITIVE PHYSICS SETTINGS

### 4.1 LEARNING DYNAMICS MODELS THAT LEVERAGE STRUCTURAL INVARIANCES

In the intuitive physics setting, reasoning at the level of objects, relations and events is a natural construct that can enable generalisation to variable scene configurations and object counts (Battaglia et al., 2016; Chang et al., 2017). We leverage this structure through the use of graph neural networks based on the Interaction Network. To improve the flexibility and generalisation capacity of the learned models, we make the following additional assumptions:

- **Distinct per-edge update functions:** Interactions between different pairs of object types are considered to be different edge types, and are encoded by edge update functions that are specific to the object pair under consideration. For example, if a ball collides into a wall, the effect of this interaction is encoded by an edge update function specific to the ball-wall interaction. If a ball rebounds off a floor, a different edge update function specific to the ball-floor interaction is used. Learning per-edge update functions specific to object pairs enables improved generalisation to various scene configurations, where these object pair interactions may be present or absent.

- **Dynamic edge activations:** In a node-dynamic graph, the set of nodes varies over time, where some nodes may be added or removed (Harary & Gupta, 1997). If a node is removed, it is assumed that edges associated with the node are also deleted. In an edge-dynamic graph, the set of edges varies over time, meaning edges may be added or removed from the graph. We represent the state of the system at time $t$ using a graph $G_t = (\mathcal{V}_t, \mathcal{E}_t)$, where $\mathcal{V}_t$ is a set of nodes and $\mathcal{E}_t$ a set of edges. $x_t^i$ is the state feature vector of node $i$, $v_t^i \in \mathcal{V}_t$, at time $t$. $d_t^{ji}$ represents the features of the edge $e_t^{ji} \in \mathcal{E}_t$ from sender node $j$ to receiver node $i$. Given that we are interested in physical contact interactions, $d_t^{ji}$ is the distance between nodes $v_t^j$ and $v_t^i$. We specify some distance cutoff $d_{cutoff}$ to determine whether an edge is active or not. If $d_t^{ji} <= d_{cutoff}$, the edge $e_t^{ji}$ is active and is added to the set of active edges $\mathcal{E}_t$. An episode can, thus, be described by a sequence of dynamic graphs $(G_0, G_1, G_2, \ldots, G_T)$ at discrete time points $0, \ldots, T$.

- **Relative information:** To capture the fact that Newtonian physics is invariant to changes in absolute object positions when their relative positions remain the same (all else being equal), we construct a dynamic coordinate system that uses the location of a reference object as its origin. The reference object is a dynamic or moving object in the scene and the positions of other objects are expressed relative to the reference object's position at each time step.

### 4.2 ACTION CLUSTERS FOR GUIDED EXPLORATION

The two main challenges in model-based reinforcement learning are: (1) model bias or inaccuracy, and (2) compounding model error over longer rollout horizons. Intuitive physics models are only approximate in nature, yet humans are able to use them to achieve flexible physical reasoning. However, for policy learning, even small model error can be catastrophic for stable policy learning (Janner et al., 2019). Allen et al. (2020) propose that humans demonstrate efficient search for physical problem solving because they leverage rich action priors, which they subsequently update as observations become available. We investigate using learned approximate models in model-based reinforcement learning in a similar way to humans. That is, we use the model to construct distributions over actions that it predicts leads to high reward. These can then be used to guide search in a reinforcement learning setting. The agent samples actions from these model-identified distributions

when exploring according to an epsilon-greedy strategy. In environments with large action spaces (particularly, continuous action spaces) or sparse rewards, the exploration problem is a difficult one (Ecoffet et al., 2021). An agent being in possession of intuitive information about actions that are promising for solving a task can direct search towards regions of the state and action spaces that provide useful signals for learning, thereby, accelerating convergence to an optimal policy.

Our approach to computing these rewarding action distributions using the learned dynamics models takes the following form:

1. For each task, we assume a set of subgoals exist that provide reward signals over shorter intervals (shorter than the full episode). The use of subgoals serves to mitigate the impact of compounding model error over longer rollout horizons, since the agent can receive signals about rewarding actions from shorter model rollouts, where predictions are likely to be more reliable (Janner et al., 2019). Additionally, the use of subgoals mitigates the credit assignment problem through decomposing the task into smaller subproblems (Sutton et al., 2000).

2. Given the initial state for a task and the first subgoal, we roll out the model over a specified horizon by randomly sampling actions. The model rollouts are performed multiple times and the actions that lead to the first subgoal being achieved as predicted by the model are stored as rewarding actions for the first subgoal. We then use K-means clustering to identify clusters of the stored rewarding actions for the first subgoal and compute their statistics (cluster means and standard deviations).

3. To limit compounding model error for the next phase of model predictions: starting from the initial state, we roll out the real environment over a specified horizon multiple times by sampling from the computed high-reward action cluster distributions (approximated as normal distributions with the cluster means and standard deviations). The last states of these real environment rollouts are then assigned as the initial states for the next set of rollouts relating to the next subgoal. For subsequent subgoals, the initial states used for model and real environment rollouts will be sampled from this constructed set of initial states.

4. The above is repeated for each subgoal until we reach the final goal.

### 4.3 Exploration Strategy for Policy Learning

We incorporate the useful action distributions computed using the learned dynamics models within a policy learning framework using an epsilon-greedy exploration strategy. Given some threshold value $\epsilon_{threshold}$ of epsilon $\epsilon$, where $0 \leq \epsilon \leq 1$: if $\epsilon <= \epsilon_{threshold}$, sample actions from the intuitive priors or cluster distributions, otherwise, sample actions according to the policy with some added noise sampled from a random process.

## 5 Experiments

### 5.1 Setup

We use the Chain Reaction Tool Environment (CREATE) developed by Jain et al. (2020). It offers a range of physics-based puzzles that require an agent to place tools in real-time to direct a ball's trajectory to reach a goal location. The action space is 2D and continuous - the $2D$ placement of the tool at regular time steps $(x, y) \in \mathbb{R}^2$, where $x, y \in [-1, 1]$. In our experiments we consider the following tool types: bucket, rotated square, box, wall and floor. The state feature vector for an object includes its $x$-velocity, $y$-velocity, $x$-position, $y$-position and a one-hot encoding representing the object type. The state observation $s_t$ corresponds to the collection of state feature vectors for objects in the scene at time $t$. We assume that an action is taken approximately every $1.33$ seconds in the environment.

Human prediction of collision resolution is inherently noisy (Smith et al., 2013). Yet even from these noisy intuitive models, humans can construct rich action priors to guide action selection in order to find an optimal solution. The 2D continuous action space and the precision of sequential tool placements required to solve tasks makes CREATE a challenging environment for exploration.

## 5.2 DYNAMICS MODELS

We learn task-agnostic dynamics models, that is, models that are not trained in the context of specific tasks. Instead, we collect observations that capture the distinct dynamics of collision interactions between a moving ball and each of the tool types under consideration (bucket, rotated square, box, wall and floor). Given a single tool in the scene, a ball is launched at various angles and velocities at the tool in order to produce observations of the tool's dynamics. We collect 1000 one-step observations for each tool type. To make the learned model invariant to absolute positions in a scene given the same relative positions of objects, we transform the observations to reflect the positions of the tools relative to the dynamic ball at each step. The dynamic ball's position is the origin of the coordinate system at each step.

For a given set of tools in a task, we learn a GNN dynamics model from the combined set of observations for the tools in the given set. We learn different edge update functions for interactions with each tool type. For example, for all tasks that involve the bucket and box tools, we use the same GNN dynamics model learned from the collated observations for the bucket and box tool types, which includes a ball-bucket edge update function and a ball-box edge update function. A dynamics model is learned for each unique combination of tool types that are included in the tasks we include in this paper. See A.1 for details on dynamics model architecture and training hyperparameters.

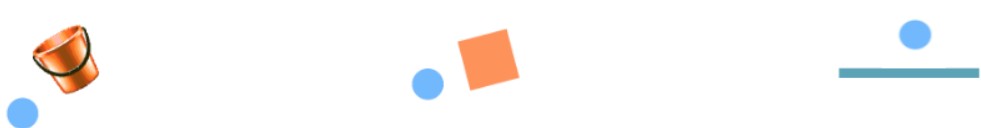

Figure 1: Individual tool dynamics observations.

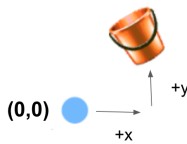

Figure 2: Dynamic relative coordinate system with moving ball's position as origin.

## 5.3 POLICY LEARNING

We use DDPG to learn deterministic policies. For each task, we compare the following two approaches:

- **DDPG with model-guided exploration (DDPG-MGE):** Combines the standard DDPG approach with guided exploration informed by the learned approximate dynamics models as described in 4.2 and 4.3. When not sampling from the learned action clusters, the exploration policy $\mu'$ used is constructed by adding noise sampled from an Ornstein-Uhlenbeck process $\mathcal{N}$ (Uhlenbeck & Ornstein, 1930) to the actor policy as done in Lillicrap et al. (2019):

$$\mu' = \mu(s_t|\theta_t^\mu) + \mathcal{N} \tag{2}$$

- **Standard DDPG (Baseline):** A standard model-free DDPG approach that uses the exploratory policy in 2.

We learn policies for the tasks shown in 3. See A.2 for details on DDPG policy and critic networks architecture and training hyperparameters.

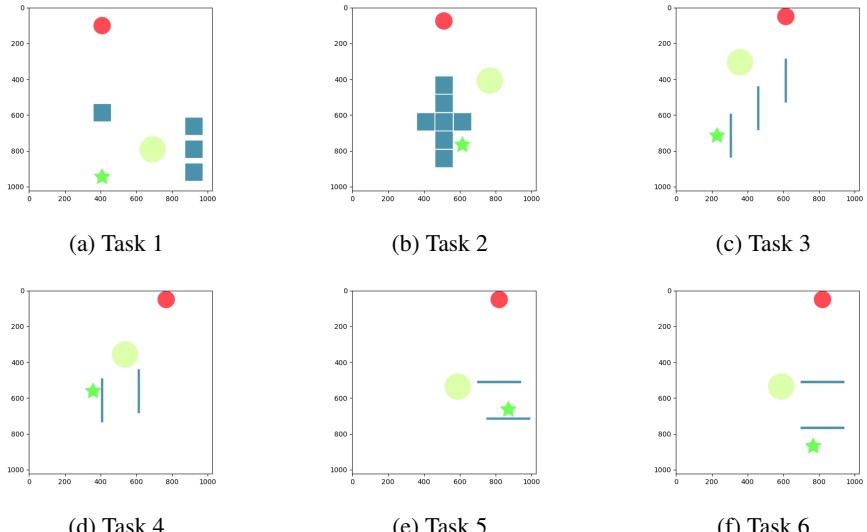

(a) Task 1       (b) Task 2       (c) Task 3

(d) Task 4       (e) Task 5       (f) Task 6

Figure 3: Tasks used in policy learning experiments.

## 5.4 RESULTS

Our experiments aim to answer the following questions:

- **Can the approximate dynamics models learned in a task-agnostic way be used to reliably identify rewarding regions of the action space across tasks?**
  Figure 4 illustrates the distributions of actions selected by sampling from the action cluster distributions identified by the learned models. Figure 5 compares the frequency of goals and subgoals being hit over 1000 episodes using model-guided exploration versus random exploration. Here, for model-guided exploration, the agent only samples from the model-identified action cluster distributions ($\epsilon_{threshold} = 1$), whereas random exploration refers to the agent sampling from its policy and adding noise sampled from an Ornstein-Uhlenbeck process. Only for Tasks 1 and 3 does random exploration lead to an occurrence of the final goal being hit. The success of model-guided exploration varies across tasks, but overall, there is a marked increase the frequency of subgoals and goals being hit under model-based exploration compared to random exploration.

- **Can exploration guided by the model be used to accelerate convergence to an optimal policy?**
  Since models are learned in a task-agnostic way, this means that in total three models were used across six tasks (a separate one for each unique combination of tool types in the scene). 2000 observations were used to train each model (that is, 2000 interactions with the real environment per model). To compute the model-identified action clusters, 200 rollouts from the real environment were required for each task. Sample efficiency gains for policy learning were achieved in four out of six tasks by using model-guided exploration (shown in Figure 6). ***Exploration epsilon*** refers to the epsilon threshold used - a threshold of 0 indicates that no model-guided exploration was used. For $\epsilon_{threshold} > 0.1$, an exploration decay strategy was employed to prevent policy learning diverging due to excessive exploration.

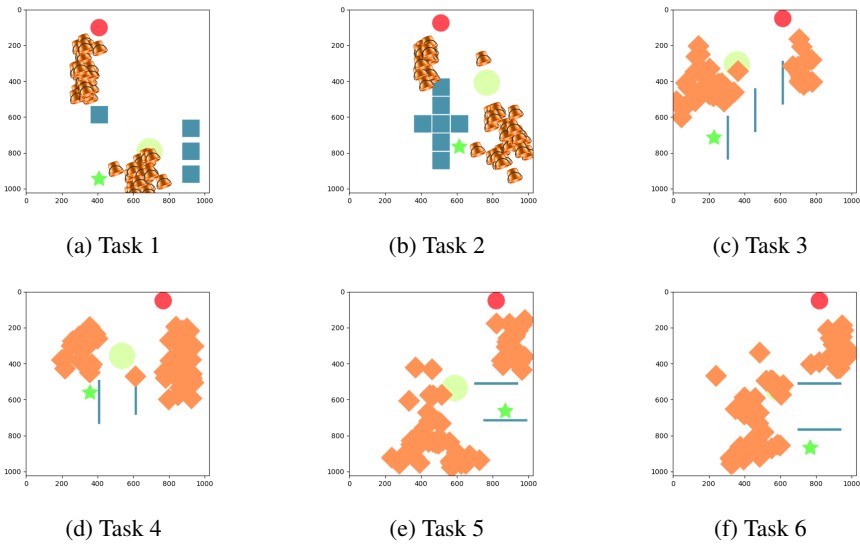

| (a) Task 1 | (b) Task 2 | (c) Task 3 |
| (d) Task 4 | (e) Task 5 | (f) Task 6 |

Figure 4: Actions sampled from cluster distributions computed using learned dynamics models.

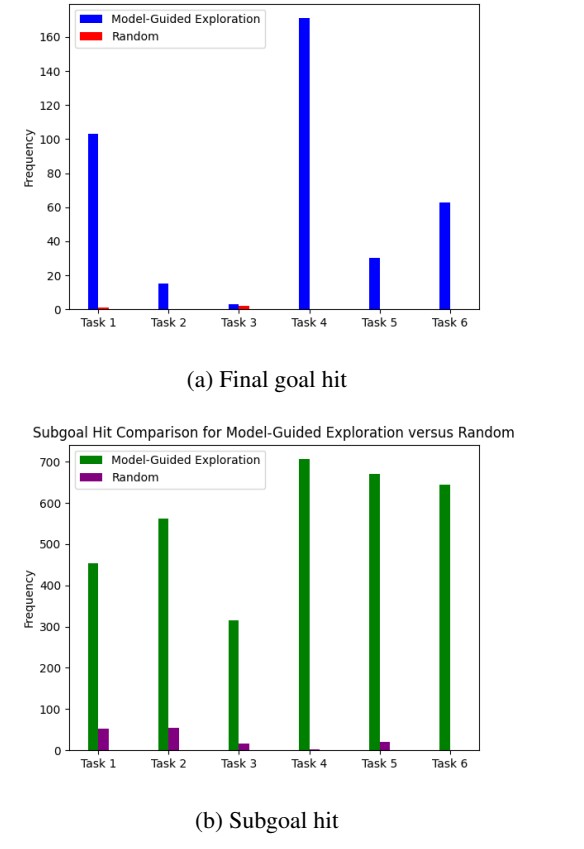

(a) Final goal hit

(b) Subgoal hit

Figure 5: Comparison of the frequency of goals and subgoals being hit over 1000 episodes using model-guided exploration versus random.

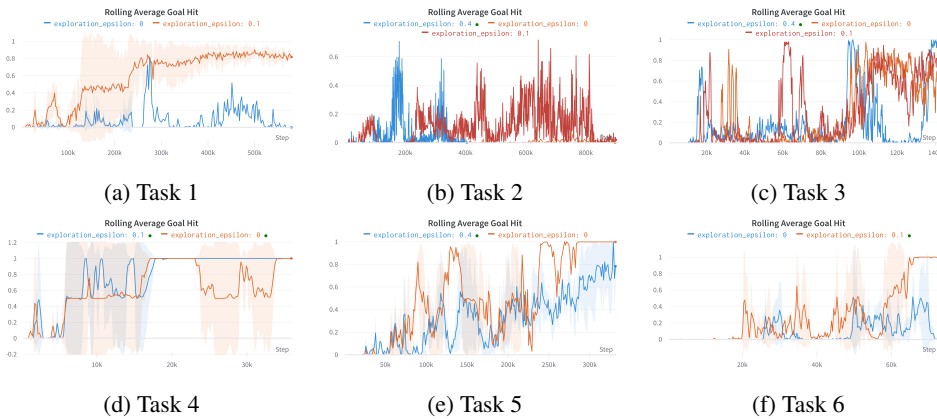

Figure 6: Cumulative reward plots at train time comparing DDPG-MGE and DDPG (baseline). Rolling average is computed over a window of 100 previous episodes and refers to the percentage of these episodes that achieved the goal.

## 6 CONCLUSIONS

Humans rely on approximate models of intuitive physics to flexibly solve a variety of physical reasoning problems. These models are inherently uncertain, yet are effectively used by humans to generalise to unseen problems. In this paper, we learn dynamics models for intuitive physics tasks using graph neural networks that explicitly incorporate the abstract structure of objects, relations and events in their design. We demonstrate that these learned models can flexibly generalise to unseen tasks and, despite being imperfect, can improve the sample efficiency of policy learning through guiding exploration to useful regions of the state and action space.

Relating to the use of imperfect or approximate models in the model-based reinforcement learning literature, there has been a focus on (1) mitigating the impact of model inaccuracies on policy learning by restricting rollout length or relying less on uncertain predictions; and (2) learning and using simplified models that abstract away details that are irrelevant for value-based planning, or, more generally, learning the model in a way that links it to its use in value-based planning. However, in environments that are challenging for exploration because of large or continuous action spaces and sparse reward signals, we propose that approximate models can be used to guide exploration towards useful regions of the action space to accelerate convergence to an optimal policy.

We show that using graph neural networks based on the Interaction Network (leveraging the abstract structure of objects, relations and events) with the additional assumptions of distinct edge update functions for unique pairs of object types, dynamic edge activations and the use of relative positional information enables effective generalisation to various scene configurations and unseen tasks. We show that these learned approximate dynamics models trained in a task-agnostic way can reliably identify rewarding regions of the action space across tasks. Sampling from distributions over model-identified rewarding actions using an epsilon-greedy exploration strategy within a reinforcement learning framework achieves sample efficiency gains in terms of accelerated policy convergence to an optimal solution.

Future work that learns subgoals for tasks instead of relying on them being specified would significantly improve our approach to identifying rewarding actions using learned dynamics models. Additionally, iteratively updating and refining the model-identified distributions over rewarding actions as real experience is collected by the agent could improve the accuracy of action selection that result in high rewards as training progresses.

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

## A  APPENDIX

### A.1  DYNAMICS MODEL NETWORK ARCHITECTURE AND HYPERPARAMETERS

The GNN consists of a set of per-edge update functions $\phi_1, \phi_2, \ldots$ for each edge type. Each edge update function is approximated using a 17-layer neural network with hidden size 64 and $tanh$ activations applied to all layers except the last. The final linear layer of the per-edge update function outputs an edge embedding vector of size 20, which is summed together with all edge embeddings for a given receiver node. The sum of the edge embeddings (size 20) is concatenated with the receiver node's feature vector and passed as an input to the per-node update function $\psi$. The per-node update function $\psi$ has the same architecture as the per-edge update function $\phi$. The node embedding vector of size 20 for the receiver node is finally passed through a 2-layer MLP, with hidden size 64 and $tanh$ activations, which outputs the predicted $x$ and $y$ positions (in the range $(-1, 1)$) for the receiver node at the next time step.

The hyperparameter values used in training the dynamics models are indicated in Table 2. These values were determined using a sweep over the set of values shown in the ***Sweep Set*** column.

Table 1: Dynamics model hyperparameters

| Hyperparameters | Values | Sweep Set |
|---|---|---|
| Number of layers | 17 | $\{5, 8, 12, 17\}$ |
| Learning Rate | 1e$-$4 | $\{\,1e-3, 1e-4, 1e-5\}$ |
| Num Epochs | 5000 | $\{\,1000, 2500, 5000, 10000\}$ |
| Batch Size | 128 | $\{\,32, 64, 128, 256\}$ |
| Optimiser | Adam | $\{\,$Adam, SGD, RMSprop$\}$ |

## A.2 DDPG: ARCHITECTURES AND HYPERPARAMETERS

The policy and critic networks each consist of 3 layers with hidden size 64.

The hyperparameter values used in training the dynamics models are indicated in Table 2. These values were determined using a sweep over the set of values shown in the *Sweep Set* column.

Table 2: DDPG hyperparameters

| Hyperparameters | Values | Sweep Set |
|---|---|---|
| Number of layers | 3 | $\{3, 6, 10\}$ |
| Policy Learning Rate | 1e$-$4 | $\{\,1e-3, 1e-4, 1e-5\}$ |
| Critic Learning Rate | 1e$-$4 | $\{\,1e-3, 1e-4, 1e-5\}$ |
| Batch Size | 128 | $\{\,32, 64, 128, 256\}$ |
| Optimiser | Adam | $\{\,$Adam, SGD, RMSprop$\}$ |
| OU $\theta$ | 0.15 | $\{\,0.1, 0.15, 0.2\}$ |
| OU $\mu$ | 0.0 | $\{\,0.0, 0.1, 0.2\}$ |
| OU $\sigma$ | 0.2 | $\{\,0.1, 0.2, 0.4\}$ |

