# OpenReview forum: "Using Approximate Models for Efficient Exploration in Reinforcement Learning"
_ICLR.cc/2024/Conference — Submitted to ICLR 2024_

### Official Review · Reviewer_SceJ · 2023-10-23

**Soundness:** 2 fair
**Presentation:** 2 fair
**Contribution:** 1 poor
**Rating:** 1
**Confidence:** 4

**Summary:**

The paper addresses the problem of using approximate models in reinforcement learning (RL). RL training is broken into several stages:
1) Model learning: learns a graph neural network model (GNN) for world dynamics, then clusters actions taken in that world model into groups predicted to lead to high rewards
2) RL learning: trains an RL model (DDPG) to learn to perform the task. Search includes an exploration condition that samples from the above action clusters as part of epsilon-greedy search.

To facilitate long horizon planning, the task is divided into a series of sub-goals that are solved sequentially. The GNN used as a world model incorporates several adjustments to apply to the particular task domain (allowing for dynamic edges, edges specific to object pairs, and encoding positions in relative coordinates). Evaluations compare against a random baseline model, showing improved probability to achieve goals in a physics-based puzzle domain.

**Strengths:**

# originality
The primary original contribution is technical: extracting clusters of promising actions leading from one sub-goal to a new sub-goal from a learned world model. The originality lies in mining the world model to bias action exploration, in lieu of the trained policy or purely random exploration.

The GNN modifications to handle these physics-based puzzles are new for this domain.

# quality
The experiments show the technique achieves a higher success rate than a random action model in achieving goals and subgoals. In most cases random actions fail to ever achieve goals, and rarely subgoals, while the method proposed achieves success on task goals with rates ranging between ~0.2% to ~16%.

# clarity
The paper compares the new method to a variety of related areas, though specific comparisons to the approach taken (separating model learning from RL training) are limited. The text is fairly clear to read.

# significance
Model-based RL has a substantial community of interest, particularly at ICLR. Providing new ways to use models for improved learning - particularly when models need not be highly accurate - has potential interest to this community.

**Weaknesses:**

# originality
The novel components of the work are minor: training GNNs to model dynamics is well-studied (as referenced) and model-based RL is a well-studied field. The technique can be viewed as a form of exploration prior to favor exploitation of high value actions. Previous work on exploration techniques may be relevant, as these span a variety of ways to leverage approximate models for altering action selection (https://lilianweng.github.io/posts/2020-06-07-exploration-drl/ provides a survey). While those techniques are typically applied to modify rewards during learning, the core approximate models are closely related to the paper technique.

# quality
In absolute terms the success rate is low for goal success (ranging from ~0.2% to ~16%). Whether this is an improvement is not clear as the experiments only compare to a baseline of random action. No comparisons are made to SOTA on the task domain, nor are other models applied to this problem. Adding baseline methods from prior work would improve the paper quality by showing evidence of improvement. Comparisons would also benefit from including metrics on the computational costs needed for different methods and their time and memory complexity, providing a full picture of the performance of the algorithm beyond the final goal outcome.

# clarity
The paper would benefit from an algorithm listing and figure demonstrating the overall training process workflow. These are left implicit in the text and make it difficult to follow the core algorithm.

# significance
Results lack non-random baselines for comparison. This makes it difficult to assess how much the new technique improves over the state of the art.

**Questions:**

- Figure 1: What is the figure depicting?
- Figure 2: This figure is probably not needed as the coordinate description is clear from the text.
- Figure 3: What are the icons in this figure?
	- What do these tasks illustrate as the task being accomplished?
- Figure 4: What do the icons in Task 1 and Task 2 mean compared to the colored blocks in the other tasks?
	- What actions do these icons indicate?
	- At what timestep(s) of the rollout?
- Figure 5:
	- Please add uncertainty estimates for the model success rates (for example, by running multiple seeds and computing the appropriate statistics). The results look clearly different, but it is hard to tell how much uncertainty there is in the models.
	- Consider adding other baseline models to compare. Random is a reasonable lower bound, but there are no other points of comparison to judge how the current work advances over other methods.
	- What performance do previous efforts on CREATE report?
- Figure 6:
	- What do the shaded areas represent?
- To strengthen the results, consider adding some ablations of the core technique.
	- For example, how does the GNN perform without distinct edge update functions for different object pairs or without dynamic edge activations?
	- This will strengthen the claims about the need for these modifications and further will clarify how much a model can be approximate and still be useful to RL learning.

---

> ### Author Response · Authors · 2023-11-20
> **Reply to Review**
>
> Thanks for your review. We found your feedback to be useful for improvement.
>
> In response to your questions:
> 1. Figure 1 depicts how a dynamic ball interacts with a single static tool (for example, bucket tool) placed in the scene with the ball being launched at various angles and velocities at the tool.
> 2. Noted.
> 3. We will work on improving this in a future iteration of the paper. In Figure 3, the goal is to place tools in the scene so that the red ball
>     hits the green star goal object. The green circle represents a subgoal.
> 4. In Tasks 1 and 2, a bucket tool was used to solve the tasks, whereas, in the other tasks, a polygon tool was used. Sampling from the
>     action clusters over all time steps across multiple episodes results in the given distribution of actions.
> 5. Noted. Thanks for the resource on model-based exploration techniques which could serve as useful baselines.
>     CREATE is an environment designed to test generalisation to new actions in reinforcement learning. As such, the original paper
>     focuses on this problem (generalisation to new, unseen tools) and does not approach the tasks as we do (where the focus is on using
>     a single tool at training and test time). There are currently no SOTA baselines for CREATE for our context.
> 6. The shaded regions represent standard deviations where multiple seeds were run.
> 7. Noted.

---

> > ### Comment · Reviewer_SceJ · 2023-11-22
> >
> > Thank you for the responses.

---

### Official Review · Reviewer_PegX · 2023-10-29

**Soundness:** 2 fair
**Presentation:** 1 poor
**Contribution:** 2 fair
**Rating:** 3
**Confidence:** 3

**Summary:**

The paper presents a graph neural network-based approach to approximately model the dynamics of intuitive physics-based tasks. Using this model, the paper proposes to identify high reward action distributions, which are then used to guide the agent’s exploration. Through physics-based puzzle games, the paper aims to show how the approach (a) learns in a task-agnostic manner and helps identify rewarding regions of the action space and (b) enables accelerated convergence by virtue of guided exploration.

**Strengths:**

The paper is mostly fairly easy to understand and aims to tackle an important issue of efficient exploration.

**Weaknesses:**

The description of the experiments needs improvement. Moreover, the results are not very convincing. Further details are described below.

**Questions:**

1.	Figures 2,3 and 4 need more descriptive captions as well as better descriptions in the text.  For example, what do the symbols correspond to in Fig 3? Similarly, in Fig 4, a legend (along with better descriptions) would have made it much easier to interpret the results.

2.	Why is it that the relative Goal hits in Fig 5a are much higher compared to other tasks?

3.	It is not immediately clear why the approach applies only to intuitive physics based tasks. I think this point needs to be emphasized better in the introduction.

4.	Over how many trials were the experiments conducted in Fig 6? In general, from the learning curves, the learning does not look stable.

5.	Doesn’t exploration epsilon=0 imply no model guidance? If so, for say, task 4, they both reach the same asymptotic performance. Why is this the case, while in some of the other tasks, the asymptotic  performances are very different?

6.	Ablations showing the effect of $\epsilon_{threshold}$ are missing.

7.	As claimed in the last line on Page 7, why would the policy learning diverge? I believe that the learning would still occur (as DDPG learns off-policy) but due to a lack of exploitation, the learning curves would not reflect the learnt policy.

8.	Perhaps the authors could have considered Phyre environments (https://ai.meta.com/tools/phyre/) to validate their approach.

9.	In Fig 6, I assumed the orange color always corresponds to exploration epsilon of 0, but for task 6, the colors are swapped.

10.	“By using…however uncertain” – This sentence towards the end of the introduction is too long and can be phrased better.

---

> ### Author Response · Authors · 2023-11-20
> **Reply to Review**
>
> Thanks for your review. We found your feedback to be useful for improvement. In response to your questions:
> 1. We will work on improving this in a future iteration of the paper. In Figure 3, the goal is to place tools in the scene so that the red ball
>     hits the green star goal object. The green circle represents a subgoal.
> 2. I assume you are referring to Task 4 here. Task 4 may be an “easier” task in the sense that there is more leeway (or more options) in
>     the exact combination of actions that lead to the goal being achieved.
> 3. Humans can use their physical reasoning intuition to focus on useful regions of the action space to improve the efficiency of learning.
>     Intuitive physics tasks also lend themselves well to generalisation - which is a key focus of the paper. Additionally, intuition is
>     approximate and imperfect, yet is still used to great effect by humans.
> 4. Learning was conducted over 2 trials due to time constraints. In a future iteration, we will increase this to improve the stability of
>     results.
> 5. We think this is because of the nature of these tasks. The agent is required to find a precise sequence of actions in a continuous
>     space and small deviations result in significant changes in reward. Some tasks may be “easier” than others in the sense that there is
>     more leeway (or more options) in the exact combination of actions that lead to the goal being achieved.
> 6. Will include in a future iteration.
> 7. Agreed.
> 8. The issue with Phyre is that it is a one step environment, which limits its applicability for reinforcement learning. Do you have
>     other suggestions for domains to test our approach?
> 9. Noted.
> 10. Noted.

---

> > ### Comment · Reviewer_PegX · 2023-11-22
> > **Thanks**
> >
> > Thanks for your responses. Perhaps the authors would find this paper useful when motivating the idea of physical reasoning: Dubey, Rachit, et al. "Investigating Human Priors for Playing Video Games." International Conference on Machine Learning. PMLR, 2018.
> >
> > I suggest increasing the number of trials in a future version, as 2 trials is too few, and may not be enough to capture the true characteristics of your approach.
> >
> > I agree that Phyre is limited because of it being a 1 step environment. An alternative could be to design custom environments using pymunk, as done in this work: https://arxiv.org/pdf/2104.08795.pdf
> >
> > I encourage the authors to continue developing and improving this work.

---

### Official Review · Reviewer_zc4U · 2023-10-30

**Soundness:** 2 fair
**Presentation:** 3 good
**Contribution:** 1 poor
**Rating:** 3
**Confidence:** 3

**Summary:**

The paper uses graph neural nets to learn the world model, then planning within the model for exploration. First pre-collect data by a random policy, then train a world model using GNN, then do planning towards subgoals (pre-defined) within the learned model, output high-rewards actions. Then exploration is performed by e-greedy. The idea is interesting, but there is still room for improvement. I would encourage authors to continually work on it. But currently, I think the paper is not ready yet for ICLR.

**Strengths:**

1. The idea is interesting, by only using the world model to guide exploration, even when the world model is very inaccurate, it will not affect the policy learning too much.

2. This model-guided exploration does seem to perform better than random exploration.

**Weaknesses:**

1. The paper assumes during the planning, a set of subgoals exist, I do think it is a strong assumption, it would be more realistic/interesting to somehow propose these sub-goals automatically.
2. Baselines are not well picked (only compared with DDPG). More should be added. For example, after the world model is learned, the policy can be learned directly within the model. But if you argue the model is not accurate enough for direct policy learning, then you should compare with it (learning a policy using data sampled from the world model). I think it would also make sense to compare with direct planning using the model.
3. Data for training the world model is pre-collected by a random policy, it works in very simple cases but wouldn’t work in more complicated tasks where you need better policy for data collection.

**Questions:**

1. The second step of computing rewarding actions is a bit like sampling-based planning, for example, CEM or MPPI. Why do you use clustering here instead of CEM or MPPI?
2. How do you sample subgoals?
3. If you first learn a model, why don’t you directly plan (for example, using MPC) in the world model, but using planning to guide the exploration, then learn a policy. Or directly do model-based reinforcement learning, since you already have a model. This combination, to me, would perform worse than planning and would be less efficient than MBRL. What’s the intuition for this combination?
4. Task descriptions are not presented, what are these tasks in Fig.3, how do they work, where are subgoals, etc.

Minior comments:
1. In section 2.2, second line, should be ‘state-action value’ instead of ‘action value’.
2. Presentation of figures is not consistent, for example, in Fig.6, some of them are with confidence intervals while some are not, some are smoothed and some are not.
3. Explanations on Fig.4 are too little, you could elaborate more on details.

---

> ### Author Response · Authors · 2023-11-20
> **Reply to Review**
>
> Thanks for your review - we found your feedback very helpful and constructive for improvement.
>
> In response to your questions:
> 1. This is an interesting question. From our understanding, sampling-based planning does not focus on learning a policy but refines the
>     distribution from which candidate actions are sampled to then be evaluated under a model?
>     Our aim was to directly assess how model inaccuracies can be mitigated through focussing on exploration priors to avoid suboptimal
>     policy learning. How do you propose we incorporate our exploration priors in a sampling-based planning setup? Or are you
>     proposing the sampling distribution acts as the exploration prior for policy learning?
> 2. The subgoals are indicated by the green circles in Figure 3. They indicate regions in space that the ball trajectory is required to pass
>     through in order to achieve the overall goal. They are hand-specified and we agree that this is a limitation. However, the focus of the
>     paper is not subgoal discovery, so perhaps a simpler heuristic algorithm to sample these subgoals would be a better approach?
> 3. Please could you elaborate on this point. What is meant by planning here? Usually planning refers to the process of using a learned
>     model to learn a policy. Are you suggesting we use the policy learned using model-generated data to guide exploration in a model-
>     free setting? We did try this and the results were poor. Because of the model inaccuracy, using standard model-based RL results in
>     poor policy learning.
>     The intuition for using exploration priors is that, like humans, an agent can benefit from exploring in useful regions of the action space
>     to obtain more informative reward signals in a sparse reward task. Humans use approximate models to solve trial-and-error intuitive
>     physics tasks.
> 4. We will work on improving this in a future iteration of the paper. In Figure 3, the goal is to place tools in the scene so that the red ball
>     hits the green star goal object. The green circle represents a subgoal.

---

> ### Comment · Reviewer_zc4U · 2023-11-21
> **Thank you for your Response**
>
> Q1: You are right, what I meant is to directly use these sampling-based planning in a model predictive control way, that is only executing the first action output by the planning, which is similar with how you use action produced by the clustering.
>
> Q3: I see your point here, nice you tried to train a policy with only using the data generated by the world model. What I meant by planning is you directly perform model predictive control within the world model, instead of train any policies. See [1,2], they first train a world model, then do planning within the world model for solving downstream tasks. Of course your work is very different with theirs, just wanna illustrate how planning could be used when having a world model.
>
> I will maintain the score I had and encourage authors continually work on it.
>
> [1]: Chua, K., Calandra, R., McAllister, R. and Levine, S., 2018. Deep reinforcement learning in a handful of trials using probabilistic dynamics models. Advances in neural information processing systems
>
> [2]: Sancaktar, C., Blaes, S. and Martius, G., 2022. Curious exploration via structured world models yields zero-shot object manipulation. Advances in Neural Information Processing Systems

---

### Official Review · Reviewer_v1eZ · 2023-11-01

**Soundness:** 2 fair
**Presentation:** 2 fair
**Contribution:** 2 fair
**Rating:** 3
**Confidence:** 4

**Summary:**

This paper focuses on utilizing learned models of the world for reinforcement learning. Planning with imagined data from an imperfect world model can lead to poor policies and bad value estimates.

To avoid issues due to model inaccuracies, this work suggests using the imagined trajectories only to guide exploration and not for updates to the policy. Concretely, model-based imagination is only used to build a prior over actions, which is used to sample actions when a non-greedy/exploratory action needs to be taken in an epsilon-greedy approach.

Experiments in an intuitive physics environment show that the proposed approach can provide benefits over standard DDPG that adds noise in parameter space to explore.

**Strengths:**

**S1.** Leveraging imperfect models of the world for RL is an important problem that interests the research community.

**S2.** The proposed approach of exploring by building action priors based on successful imagined trajectories is an interesting idea.

**S3.** Most of the paper is well-written and easy to follow.

**Weaknesses:**

**W1.** A key weakness of the present submission is in the empirical evaluation.

It is not clear if planning with the learned model (using the world model to train DDPG) would actually be problematic in this setting, which was posed as a vital motivating factor for this work.

Further, the paper only compares the proposed approach to standard DDPG. However, many approaches to model-based exploration use a learned model solely for exploration (see [1, 2]). E.g., curiosity-driven approaches use a learned model to derive intrinsic rewards based on the model’s prediction error, information gain, etc. The paper would benefit from discussions and empirical comparisons to other model-based approaches to exploration.
For instance, it is unclear if model inaccuracies would impact curiosity-based exploration more than the approach presented here.

Also see Q2.


**W2.** The paper makes some strong assumptions that limit the generality and applicability of the proposed approach.

One strong assumption that the paper makes is to have a resettable ‘true’ environment, which allows multiple environment rollouts from the same environment state (point 3 on page 5). Another crucial assumption is the availability of subgoals in sparse reward environments.

On a minor note, Figure 6 could be improved as it is currently hard to analyse. It would be better to use the same colors to denote approaches across the sub-figures.

—------------------—------------------—------------------—------------------—------------------

### References

[1] Amin, S., Gomrokchi, M., Satija, H., van Hoof, H., & Precup, D. (2021). A survey of exploration methods in reinforcement learning. arXiv preprint arXiv:2109.00157

[2] Moerland, T. M., Broekens, J., Plaat, A., & Jonker, C. M. (2023). Model-based reinforcement learning: A survey. Foundations and Trends® in Machine Learning

**Questions:**

Q1. What are the subgoals in the tasks used for evaluation?

Q2. While the proposed idea shows some benefit over standard DDPG, the learning dynamics appear quite unstable in Figure 6. Can the authors explain why this might be the case?

---

> ### Author Response · Authors · 2023-11-20
> **Reply to Review**
>
> Thank you for the helpful review. We appreciate the interesting ideas and points you raised in your feedback.
>
> Firstly, in response to your questions:
> The subgoals are indicated by the green circles in the tasks - see Figure 3.
> The learning dynamics appear unstable because of the nature of these tasks. The agent is required to find a precise sequence of actions in a continuous space and small deviations result in significant changes in reward.
>
> We did find that planning with the world model results in significantly deteriorated performance - we will include these results in a future iteration of the paper. Your suggestion to include comparisons with other model-based exploration methods (for example, curiosity-driven approaches) is a very useful one and thanks for sharing the papers.

---

> > ### Comment · Reviewer_v1eZ · 2023-11-21
> >
> > Thank you for your response and clarifications. I hope the feedback from the reveiwers can help you with your future revisions.

---

### Meta-Review · Area_Chair_Bu9c · 2023-12-11

**Metareview:**

This paper presents a way to use intuitive models to enhance exploration in RL. The paper presents an intuitive physics model using graph neural networks to help accelerate the convergence to an optimal policy. As noted by reviewers, this paper needs experiments demonstrating the proposed approach's benefits or trade-offs. In line with the reviewers' assessment, I do not recommend this paper for acceptance.

**Justification For Why Not Higher Score:**

This paper is missing evidence that the method is addressing a key problem.

**Justification For Why Not Lower Score:**

N/A

---

### Decision · Program_Chairs · 2024-01-16

Reject